1 **Predict the Risk of Dyslipidemia via Deep Neural Networks for Survival Data**

2

3 Hailun Liang [a#], Dongzuo Liang [b#], Lei Tao [c], Xiaoshuai Zhang [d], Xiao Li [e], Da Zheng [f], Yuhan

4 Chen [a], Meixi Yi [a], Fang Tang [g h*]

5

6 [a] School of Public Administration and Policy, Renmin University of China, Beijing, China

7 [b] School of Statistics, Renmin University of China, Beijing, China

8 [c] Department of Public Policy, City University of Hong Kong, Hongkong, China

9 [d] Department of Data Science, School of Statistics, Shandong University of Finance and

10 Economics, Jinan, China

11 [e] Department of Clinical Pharmacy, The First Affiliated Hospital of Shandong First Medical

12 University & Shandong Provincial Qianfoshan Hospital

13 [f] Department of Computer Science, Johns Hopkins University

14 [g] Center for Big Data Research in Health and Medicine, the First Affiliated Hospital of Shandong

15 First Medical University & Shandong Provincial Qianfoshan Hospital, Jinan 250014, PR China

16 [h] Shandong Provincial Qianfoshan Hospital, cheeloo College of Medicine, Shandong University,

17 Jinan 250014, PR China

18

19 # The authors contributed equally.

20 *Corresponding author: Fang Tang, Center for Big Data Research in Health and Medicine, The

21 First Affiliated Hospital of Shandong First Medical University & Shandong Provincial

22 Qianfoshan Hospital, Jingshi Road 16766, 250014, Jinan, China. E-mail:

23 tangfangsdu@gmail.com ORCID： 0000-0002-4378-594X

**Abstract**

**Background:** Dyslipidemia is an important risk factor for coronary artery disease and stroke. Early detection and prevention of dyslipidemia can markedly alter cardiovascular morbidity and mortality. Cox proportional hazard model has been commonly employed for survival datasets to construct the prediction model. Recently, the data-driven learning algorithm began to be used to analyze right-censored survival data. However, there is no attempt to use deep neural networks in dyslipidemia prediction. The objective of this study is to predict the risk of dyslipidemia via deep neural networks for survival data.

**Methods:** The study cohort was based on the routine health check-up data from 6,328 participants aged 19 to 90 years and free of dyslipidemia at baseline. A deep neural network (DNN) was used to develop risk models for predicting dyslipidemia. Cox Proportional Hazards (Cox) and Random Survival Forests (RSF) were applied in comparison with the DNN model. As metric of performance, we use the time-dependent concordance index ($C^{td}$-index).

**Results:** The $C^{td}$-index of the prediction models by using DNN was 0.802. The DNN model performed significantly better than Cox and RSF model ($C^{td}$-index: 0.735 and 0.770, respectively). The improvement of DNN over the competing methods was statistically significant. Moreover, DNN provides performance gain on time intervals compared to conventional survival models.

**Conclusions:** DNN is a promising method in learning the estimated distribution of survival time and event while capturing the right-censored nature inherent in survival data. DNN achieves large and statistically significant performance improvements over previous intuitive regression model and state-of-the-art data-mining methods.

**Key Words:** dyslipidemia, risk prediction, deep neural network, survival analysis

47    **New & Noteworthy:**

48    ● This study applies a DNN based learning algorithm, to develop a risk prediction model for

49    dyslipidemia based on routine check-up data.

50    ● DNN provides performance improvements measured by the C-index score over the COX

51    regression model and RSF in dyslipidemia prediction.

52    ● The DNN model may provide a feasible and accurate approach for identifying the high-risk

53    population among undiagnosed dyslipidemia subjects based on their routine check-up data.

54

55    **1. Background**

56    Dyslipidemia is the metabolic abnormality of lipoprotein in the human body, mainly including

57    the increase of total cholesterol and low-density lipoprotein cholesterol, triglyceride, and decreased

58    high-density lipoprotein cholesterol, etc [1]. Unhealthy lifestyles, such as high cholesterol diet, an

59    inactive lifestyle, and smoking, are particularly high-risk factors in developing dyslipidemia. In

60    China, the prevalence of dyslipidemia is rapidly increasing. One recent study has illustrated that

61    the incidence density of dyslipidemia in China is as high as 101/1000, and 121/1000 for men and

62    69/1000 for women, respectively [2]. Dyslipidemia is also a major risk factor for cardiovascular

63    diseases (CVD), a serious threat to people's health, especially in developing countries. The study

64    shows that CVD mortality increased by 41% between 1990 and 2013, mainly due to low and lower-

65    middle-income countries [3]. Moreover, the burden of dyslipidemia and CVD is now growing

66    faster than our ability to combat. Considering the increased burden caused by dyslipidemia, it is

67    of great significance to manage the disease by early predicting, detecting, and dealing with risk

68    factors [4,5,6].

69       Traditionally, predictive models were used as the public health responses to disease control.

70    Statistical models and epidemiological models have been commonly employed to construct the

71    predictions. In terms of the methods and statistics of identifying and predicting risk factors, Cox

72    proportional hazard model (CPH) has been widely used. CPH relates the log of the hazard ratio to

73    a linear function of the predictors [7], making it easy to model cause-specific hazards [8]. However,

74    CPH has some limitations. For example, the validity of its results will be affected by factors such

75    as modeling and proportional risk assumption [5]. And CPH has been proven to have a high

76    variance if the model is greatly complex [8]. In order to deal with a variety of potential results, it

77    is necessary to apply appropriate methods to consider and manage competing risks [9].

78    Furthermore, dyslipidemia possibly also lead to some complications such as atherosclerosis,

79    coronary heart disease (CHD), peripheral artery disease (PAD), stroke, and others, which may

80    delay or mask the symptoms of dyslipidemia [1], making it more difficult to use CPH for accurate

81    prediction. Consequently, it is crucial to use a high prediction capacity method in a complex

82    situation.

83       Random Survival Forests (RSF) is a nonparametric algorithm, which has been developed to

84    surmount the unsolvable problems of the Cox and other classical models. RSF can cope with plenty

85    of covariates and the correlation between the response and the predictors [6,10]. In addition, RSF

86    can also be applied to select or rank variables, making it to achieve successful risk predictions for

87    several diseases [3]. Meanwhile, to improve the accuracy of disease prediction for survival data,

88    Deep Neural Networks (DNN) have been applied in the field of precise prevention, which is some

89    of the most prominent non-linear algorithms [11]. Recently, it has been suggested that DNN could

90    be a good model for biological networks due to some near-human performance [12].

91      However, these models' performance differs significantly depending on the assumptions and

92      values of parameters they employ. In addition, some of these models also tend to simplify the

93      complex biological and social processes in which real diseases involve. For these reasons, it is best

94      not to depend on a specific projection coming from a single model. Using multiple models and

95      updating approaches can help diminish some of the limitations inherent in modeling. Recently, a

96      data-driven learning algorithm began to be used to analyze spatial-temporal data. However, there

97      is no attempt to use DNN to predict dyslipidemia. Therefore, the goal of this study is to apply the

98      DNN prediction model to predict the risk of dyslipidemia. To observe the performance gain of our

99      model, we also compare the predictive power of the DNN model with CPH and RSF.

100

101    **2. Materials and Methods**

102    2.1. Subjects

103      We conducted a prospective cohort study of 6,328 participants who received routine health

104    check-up at Shandong Provincial Qianfoshan Hospital. These participants met the following

105    criteria: (1) aged between 19 and 90 years; (2) received their first check-ups between 2010 and

106    2015; (3) received at least three health checks during the 5-year follow-up; (4) individuals who

107    had been diagnosed as having dyslipidemia, diabetes, cardiovascular disease, hepatosis, renal

108    dysfunction, or hypothyroidism at baseline were excluded. The study was approved by the

109    Institutional Review Board of Shandong Provincial Qianfoshan Hospital. The study was conducted

110    in accordance with the principles of the Declaration of Helsinki. The written informed consent was

111    obtained from all eligible participants.

112

113    2.2. Outcome and predictor variables

The prediction outcome of this study was the probability of developing dyslipidemia. We defined dyslipidemia according to the 2016 Chinese guidelines for the management of dyslipidemia in adults [13]. Dyslipidemia was defined as having triglycerides (TG) $\geq$ 2.3mmol/L, and/or low-density lipoprotein cholesterol (LDL-C) $\geq$ 4.1 mmol/L, and/or total cholesterol (TC) $\geq$ 6.2 mmol/L, and/or high-density lipoprotein cholesterol (HDL-C) $\leq$ 1.0 mmol/L.

We employed the predictor variable set collected from the anthropometric and laboratory tests. These predictor variables are closely related to the risk of developing dyslipidemia and can be available in the clinical practice, facilitating the deployment of the model. The anthropometric variables included height, weight, BMI, systolic blood pressure (SBP), and diastolic blood pressure (DBP). In terms of laboratory biomarkers, peripheral blood samples were collected after an overnight fast for measuring the following variables: absolute lymphocyte count (ALC), alanine transaminase (ALT), absolute monocytes count (AMC), aspartate transaminase (ASTblood urea nitrogen (BUN), blood uric acid (BUA), fasting blood-glucose (FBG), gamma-glutamyl transpeptidase (GGT), neutrophil granulocyte (GRA), hematocrit (HCT), highdensity lipoprotein cholesterol (HDL-C), hemoglobin (HGB), low-density lipoprotein cholesterol (LDL-C), mean corpuscular hemoglobin (MCH), mean platelet volume (MPV), platelet large cell ratio (P-LCR), red blood cell count (RBC), serum creatinine (SCr), total cholesterol (TC), triglycerides (TG), and white blood count (WBC).

2.3. Prediction Models

2.3.1. Cox proportional hazard model

Cox proportional hazards model (CPH) is the most widely-used statistical model in the medical setting for investigating the association between the survival time of patients and one or more

137    predictor variables [14]. It has been commonly used in a cohort study to identify the risk factors

138    and construct the prediction model using time-to-event data [15]. However, CPH subjects to

139    restriction about the underlying stochastic process, which assumes the hazard rate are constant and

140    the log of the hazard rate is a linear function of covariates. CPH suffers from high variance when

141    the model is complicated, and nonlinear effects exist.

142

143    2.3.2. Random Survival Forest

144        Random survival forest (RSF) is a data-driven learning algorithm that can automatically deal

145    with the nonlinear effects and interactions among the predictors. Similar to the random forests (RF)

146    [16], RSF uses bootstrap method to randomly select samples from the dataset to construct survival

147    tree models and uses 37% out-of-bag data from each sample to calculate model accuracy [10].

148    While difference between RSF and RF lies in that response variable in RSF is a survival time,

149    implicating that the data might be censored.  In addition, RSF and RF differs in that RSF splits the

150    data at the node with the criterion that maximizes the survival difference. Therefore, RSF is

151    specifically suitable for right-censored data to build prediction model to study the complicated

152    relationship between various predictors and response. It can be used for event-specific selection of

153    risk factors in a nonparametric way with no restrictive assumption; thus, it is suitable to reduce the

154    data dimension of highly correlated biomarker data that are linked with event time of interest [17].

155    RSF has been applied to identify risk factors for several diseases.  An RSF is a collection of

156    randomly grown survival trees, which are generally grown very deeply with many terminal nodes.

157    By using random feature selection at each node, each tree is grown using an independent bootstrap

158    sample of the learning data. The splitting rules are either event-specific or combine event-specific

159    splitting rules across the events [17,18].

160

### 2.3.3. Deep Neural Network

The problem of survival analysis has also received substantial recent attention in the deep learning literature. Recently, several works applied deep neural networks (DNNs) to learn complex representations of risk and capture the time-dependent influence of covariates on survival. The current study applied a DNN model, called DeepHit and developed by Lee et al., [19] to learn the estimated distribution of survival time and event while capturing the right-censored nature inherent in survival data. This DNN model makes no assumptions about the underlying stochastic process, making it possible to smoothly learn the nonlinear relationship between the variables and the disease risks.

We treated participants' survival time as discrete and the time frame as finite. The time set was $T = \{0, \ldots, T_{max}\}$ for a maximum time horizon $T_{max}$. We assumed that exactly one event eventually occurs for each participant and considered one event of interest. The current DNN model employed a network architecture that consists of multiple fully-connected layers and a softmax layer as the output layer. The model was trained by using a loss function that exploits both survival times and relative risks [19]. Figure 1 showed the architecture of the current DNN model.

( Figure 1 insert here )

### 2.4. Analysis

We assessed the baseline characteristics of participants with and without incident dyslipidemia by using a t-test for continuous variables and a chi-square test for categorical variables. The DNN model was used to develop risk models for predicting dyslipidemia. CPH and RSF models were

183     applied in comparison with the DNN model. To evaluate the prediction performance of the three

184     models, we randomly separated the data into training set (80%) and testing set (20%). As our

185     metric of performance, we use the time-dependent concordance index ($C^{td}$-index) [20]. The

186     concordance index measures the extent to which the ordering of survival times of pairs agrees with

187     the ordering of their predicted risk, which is a widely-used metric for evaluating the performance

188     of survival models [21].

189

190     **3. Results**

191     3.1. Descriptive statistics

192       The baseline characteristics of the study cohort by gender were summarized in Table 1. A total

193     of 2219 dyslipidemia participants were included in this study. Male participants (41.9%) were

194     more likely to develop dyslipidemia than females (21.49%). The mean age of males and females

195     was 45.2 and 41.4, respectively. There were no significant differences in TC, MPV, and P-LCR

196     between the males and females.  Except for these three variables, the differences of remaining

197     variables between the patient groups were statistically significant.

198

199                          ( Table 1 insert here )

200

201       Kaplan-Meier survival estimates comparing males and females were visualized in Figure 2.

202     We observed a significant difference between male and female participants, with higher survival

203     probabilities for females than males over time. Therefore, the prediction models were respectively

204     constructed by males and females.

205

206 

207

208

209 3.2. Risk model with Cox

210     Table 2 shows the results of the Cox prediction model for dyslipidemia based on full-samples.

211 The significant variables included age, BMI, BUN, GGT, GRA, HDL-C, LDL-C, TC, P-LCR, and

212 TG. Except for HDL-C and P-LCR, other significant variables are positive factors in predicting

213 dyslipidemia. BUA was a non-predictive variable in predicting dyslipidemia.

214

215 

216

217     We further conducted the Cox prediction model for males and females in Figure 3, respectively.

218 For males, the most predictive variables included HDL-C, TG, TC, and LDL-C. Four variables

219 were found to be irrelevant for predicting dyslipidemia in males, including WBC, P-LCR, AMC,

220 ALC. For females, LDL-C, ALC, GRA, and TG are the most influential factors in predicting

221 dyslipidemia.

222

223 

224

225 3.3. Comparisons of the performance in Cox, RSF, and DNN models

226     Table 3 compared the prediction performance of Cox, RSF, and DNN models using $C^{td}$-index

227 values based on testing set and also by gender over time. The $C^{td}$-index of the DNN model at the

228 25th percentile survival time was 0.802. The DNN model performed significantly better than Cox

229 and RSF model ($C^{td}$-index: 0.735 and 0.770, respectively). Similarly, at the 50th and 75th percentile

230 survival time, the DNN model showed higher predictive power than Cox and RSF models.

231 Comparing the prediction performance of these models by gender, we also found DNN model

232 achieved the highest performance than Cox and RSF models. In addition, our results showed that

233 the DNN model significantly provided a better prediction of dyslipidemia for females than males.

234

235 ( Table 3 insert here )

236

237     Figure 4 visualized the predictive power of Cox, RSF, and DNN models over time based on

238 testing set. Compared to Cox models and RSF, the DNN model shows a significantly higher

239 performance over time. We further visualized the performance of the three models. For males,

240 RSF exhibited a very similar predictive power with the Cox model, whereas our DNN model still

241 largely outperformed the Cox and RSF. For females, at the beginning of the prediction time, no

242 significant differences were observed in performance between Cox and RSF, while with the time

243 evolved, RSF showed a performance improvement over Cox. DNN model always outperformed

244 the Cox and RSF.

245

246 ( Figure 4 insert here )

247

248     In sum, DNN showed the highest predictive power and provided performance improvements

249 in dyslipidemia prediction over Cox and RSF in this study.

250

251 **4. Discussion**

252   This study aimed to apply a DNN based learning algorithm, DeepHit, to develop a more

253   accurate risk prediction model for dyslipidemia. Several common predictors were extracted from

254   the routine health check-up data to construct the prediction model. Our results showed that DNN

255   fitted our data well and provided performance improvements measured by C-index score over the

256   intuitive regression model and state-of-the-art data-mining methods in dyslipidemia prediction.

257   We compared the predictive power of DNN with CPH and RSF and found DNN was a superior

258   model in dyslipidemia prediction. The improved accuracy on dyslipidemia prediction of the

259   DeepHit model could be attributed to its flexible processing ability, which can smoothly learn the

260   nonlinear relationship between the variables and the disease risks [19]. The Cox regression model

261   [14] follows a strict assumption that the underlying relationship between variables and the hazard

262   rate is a linear function. However, the prevalence of dyslipidemia is rather complicated and beyond

263   the applicable conditions of the Cox Model, which might significantly limit the accuracy of its

264   predictive power. Although RSF [17] is a data-driven algorithm that can automatically learn the

265   underlying patterns between the covariates and the risk events, it has a limited predictive capacity,

266   particularly in the presence of many covariates. Thus, our DeepHit model, which makes no

267   assumptions about the underlying stochastic process, can provide a more accurate prediction of

268   dyslipidemia events.

269   Our findings on the performance improvement of DNN in dyslipidemia prediction were

270   consistent with existing literature, illustrating that DNN models were powerful techniques for

271   disease prediction. For example, Zhao and Feng [22] found that the DNN model better performed

272   than existing methods, such as a standard CPH and Cox-nnet model, in predicting the development

273   and progression of cardiovascular disease among older populations. Lee et al. [23] presented a new

274   DNN model for overcoming the major disadvantages of the CPH in predicting non-small cell lung

275  cancer patients' recurrence probabilities after surgery; results demonstrated that semi-

276  unsupervised binned-time survival analysis (su-DeepBTS) model exhibited the best performance

277  with a concordance index(C-index) of 0.7306 and an area under the curve (AUC) of 0.7677, better

278  than supervised binned-time survival analysis (s-DeepBTS) and CPH (C-index of 0.7048 and

279  0.7126 and AUCs of 0.7390 and 0.7420 respectively). For better predicting the progression from

280  mild cognitive impairment(MCI) to Alzheimer's disease(AD), Sebastian Pölsterl et al. [24]

281  proposed a wide and deep neural network model that fused information of anatomical shape and

282  tabular clinical data from survival analysis. Their study indicated that this model was superior to

283  a baseline neural network on shapes and a linear model on common clinical biomarkers, which

284  both enhanced clinical variables and improved prediction performance.

285      DeepHit was developed in the late 2018 as a new deep learning approach to survival analysis.

286  Although DeepHit exhibited high predictive power over previous intuitive regression model and

287  state-of-the-art data-mining methods, up to present few studies have applied it to disease risk

288  prediction. Except for two studies applying it to predict breast cancer [19] and cystic fibrosis [25],

289  no applications has been developed to predict dyslipidemia risks. Therefore, to the best of our

290  knowledge, this was the first application of the DNN model in predicting dyslipidemia based on

291  the routine check-up data. We specifically confirmed that DNN was a useful tool in dyslipidemia

292  risk prediction. In addition, given the high predictive power of DNN compared to other existing

293  models, our research contributes to the current literature by indicating that nonlinear relationships

294  between predictors and survival times are crucial for assessing the risk of dyslipidemia. Predictive

295  models based on a linear assumption may limit the accuracy of their predictions and hinder

296  practitioners' ability to precisely evaluate the dyslipidemia risks of their patients. Thus, practically,

297  our DNN predictive model provides a feasible and accurate approach for identifying the high-risk

298  population among undiagnosed dyslipidemia subjects based on their routine check-up data. Given

299  the rapidly increasing prevalence of dyslipidemia in China [26], identifying the individual risk of

300  dyslipidemia carries significant implications for early intervention strategies. In addition,

301  dyslipidemia is an essential risk factor for a variety of diseases, such as cardiovascular disease,

302  heart disease, and stroke [27,28]. The policy intervention plans against the prevalence of

303  dyslipidemia will undoubtedly reduce the risk of those chronic diseases among the Chinese

304  population.

305  Despite these strengths, our research has some limitations. Firstly, our samples consist of

306  patients from large medical institutions with high socioeconomic status, limiting the robustness of

307  the model. It thus should be cautious about generalizing our findings to other groups with distinct

308  geographic and socioeconomic features. Further validation utilizing other data sources,

309  particularly a nationally representative dataset, could make the predictive power of the DNN model

310  in dyslipidemia more accurate. Secondly, we excluded the patients who had already had

311  dyslipidemia at the baseline, which might lead us to underestimate the actual survival time. Finally,

312  our predicators of dyslipidemia were all from the routine check-up. Further work could consider

313  possible environmental variables and other genetic-related factors.

314

315  **5.Conclusion**

316  In conclusion, our research confirms that DNN approaches are powerful tools to identify

317  subjects with a high risk of dyslipidemia. In addition, our DNN significantly outperformed the

318  other two models in predicting dyslipidemia for survival data. Based on our research, a more

319  precise assessment can be performed in the health populations with DNN to guide the early

320  classification of risks and thus effectively lower the incidence of dyslipidemia and other-related

321 disease.

322

## Authors' contributions

324 Conception and design: Hailun Lian, Xiaoshuai Zhang, Xiao Li, Fang Tang

325 Data analysis and interpretation: Hailun Lian, Dongzuo Liang, Lei Tao, Yuhan Chen, Meixi Yi,

326 Da Zheng

327 Manuscript writing: Hailun Lian, Dongzuo Liang, Lei Tao, Xiaoshuai Zhang, Xiao Li, Yuhan

328 Chen, Meixi Yi, Fang Tang, Da zheng

329 Final approval of manuscript: Hailun Lian, Fang Tang, Xiaoshuai Zhang, Xiao Li, Fang Tang

330

## Acknowledgments

332 This study was supported by the National Natural Science Foundation of China (No. 71804183

333 and 71804093)

334

## Funding

336 National Natural Science Foundation of China (No. 71804183 and 71804093)

337

## Statement on conflicts of interest

339 We declare that we have no financial and personal relationships with other people or organizations

340 that can inappropriately influence our work, there is no professional or other personal interest of

341 any nature or kind in any product, service and/or company that could be construed as influencing

342 the position presented in, or the review of, the manuscript entitled"Predict the Risk of Dyslipidemia

343 via Deep Neural Networks for Survival Data".

344

**Ethics approval and consent to participate**

The study was approved by the Institutional Review Board of Beijing Physical Examination Center. The study was conducted in accordance with the principles of the Declaration of Helsinki. The written informed consent was obtained from all eligible participants.

**Availability of data and materials**

The data used in this study is provided by Shandong Provincial Qianfoshan hospital. We have obtained the right to use the data through application. If anyone wants to obtain data from this study, they can contact Fang Tang (Contact information: Center for Big Data Research in Health and Medicine, The First Affiliated Hospital of Shandong First Medical University & Shandong Provincial Qianfoshan Hospital, Jingshi Road 16766, 250014, Jinan, China. E-mail: tangfangsdu@gmail.com)

**Consent for publication**

Not applicable

**Summary points**

**What was already known on the topic:**

- Early detection and prevention of dyslipidemia can markedly alter cardiovascular morbidity and mortality.

- Cox proportional hazard model (CPH) and Random Survival Forests (RSF) are two common tools to construct the predictive model for dyslipidemia.

367  ● The existing models usually have poor performance in dyslipidemia prediction as they strictly

368    follow basic assumptions and values of parameters.

369

370  **What this study added to our knowledge:**

371  ● This study applies a DNN based learning algorithm, DeepHit, to develop a risk prediction

372    model for dyslipidemia based on routine check-up data.

373  ● DNN provides performance improvements measured by the C-index score over the COX

374    regression model and RSF in dyslipidemia prediction.

375  ● The DNN model may provide a feasible and accurate approach for identifying the high-risk

376    population among undiagnosed dyslipidemia subjects based on their routine check-up data.

377

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

Table 1 Baseline Characteristics by Gender

| Variables | Male | Female | P-value |
|---|---|---|---|
| Dyslipidemia | 1628(41.90%) | 591(24.19%) | <0.001 |
| Age | 45.221(±14.414) | 41.357(±12.53) | <0.001 |
| ALC | 2.245(±0.586) | 2.081(±0.528) | <0.001 |
| BUN | 5.42(±1.202) | 4.585(±1.095) | <0.001 |
| AMC | 0.352(±0.111) | 0.304(±0.094) | <0.001 |
| TC | 4.704(±0.683) | 4.7(±0.699) | 0.814 |
| HDL-C | 1.561(±0.299) | 1.791(±0.327) | <0.001 |
| ALT | 23.613(±17.556) | 16.368(±16.167) | <0.001 |
| AST | 21.221(±9.166) | 18.868(±7.814) | <0.001 |
| RBC | 4.993(±0.357) | 4.413(±0.309) | <0.001 |
| HCT | 0.449(±0.027) | 0.392(±0.027) | <0.001 |
| SCr | 76.593(±10.769) | 56.355(±8.47) | <0.001 |
| BUA | 351.888(±70.862) | 257.086(±53.686) | <0.001 |
| MCH | 30.771(±1.447) | 29.589(±2.193) | <0.001 |
| MPV | 10.464(±0.794) | 10.47(±0.807) | 0.791 |
| FBG | 5.23(±0.627) | 4.999(±0.481) | <0.001 |
| HGB | 153.379(±9.864) | 130.336(±10.848) | <0.001 |
| GRA | 3.35(±1.074) | 3.2(±1.093) | <0.001 |
| TG | 1.081(±0.341) | 0.879(±0.334) | <0.001 |
| LDL-C | 2.648(±0.525) | 2.427(±0.583) | <0.001 |
| WBC | 6.123(±1.427) | 5.723(±1.394) | <0.001 |
| P-LCR | 28.487(±6.617) | 28.464(±6.708) | 0.896 |
| GGT | 28.595(±22.144) | 15.883(±14.402) | <0.001 |
| SBP | 129.361(±17.22) | 118.84(±17.119) | <0.001 |
| DBP | 82.267(±11.004) | 73.586(±10.048) | <0.001 |
| BMI | 24.589(±3.105) | 22.25(±3.063) | <0.001 |

472
473                          Table 2 Cox Proportional Hazard Model for Predicting Dyslipidemia

| Variables | Coe | Z statistic | P-value | HR | Lower | Upper |
|---|---|---|---|---|---|---|
| Age | 0.004 | 2.080 | 0.038 | 1.004 | 1.000 | 1.007 |
| BMI | 0.023 | 2.758 | 0.006 | 1.023 | 1.007 | 1.040 |
| BUA | 0.001 | 1.477 | 0.140 | 1.001 | 1.000 | 1.001 |
| BUN | 0.037 | 1.812 | 0.070 | 1.038 | 0.997 | 1.081 |
| GGT | 0.004 | 3.859 | 0.000 | 1.004 | 1.002 | 1.006 |
| GRA | 0.047 | 2.039 | 0.041 | 1.048 | 1.002 | 1.096 |
| HDL-C | -1.437 | -9.686 | <0.00001 | 0.238 | 0.178 | 0.318 |
| LDL-C | 0.519 | 3.995 | <0.0001 | 1.681 | 1.303 | 2.169 |
| P-LCR | -0.006 | -1.711 | 0.087 | 0.994 | 0.987 | 1.001 |
| TC | 0.571 | 5.462 | <0.00001 | 1.769 | 1.442 | 2.171 |
| TG | 0.642 | 7.656 | <0.00001 | 1.901 | 1.613 | 2.241 |

474
475
476
477
478
479
480
481
482
483
484
485
486
487
488
489
490
491
492
493
494
495
496
497
498
499

Table 3 $C^{td}$-index for Prediction Model with Cox, RSF and DNN Model

| Models | 25% | 50% | 75% |
|---|---|---|---|
| **Full testing set** | | | |
| Cox | 0.735 | 0.735 | 0.732 |
| RSF | 0.770 | 0.768 | 0.766 |
| DeepHit | 0.802 | 0.798 | 0.794 |
| **Testing set of male** | | | |
| Cox | 0.733 | 0.727 | 0.726 |
| RSF | 0.744 | 0.738 | 0.737 |
| DeepHit | 0.831 | 0.804 | 0.809 |
| **Testing set of female** | | | |
| Cox | 0.771 | 0.786 | 0.780 |
| RSF | 0.812 | 0.818 | 0.807 |
| DeepHit | 0.851 | 0.849 | 0.834 |

500

501
502

503

504

505

506

507

508

509

510

511

512

513

514 **Figure Legend**

515 **Figure 1** The architecture of the DNN model

516 **Figure 2** Kaplan-Meier survival estimates comparing male with female participants

517 **Figure 3** Cox Proportional Hazard Model for Predicting Dyslipidemia in Male (A) and Female

518 (B)

519 **Figure 4** Comparison of Ctd-index performance in Cox, RSF and DNN Models based on full

520 testing set (A); by the male (B) and female (C)

521

522

523

524

525

526

527

528

529

530

531

532

533

534

535

$$\Pr\{T = t_j\}, t_j \in \mathcal{T}$$

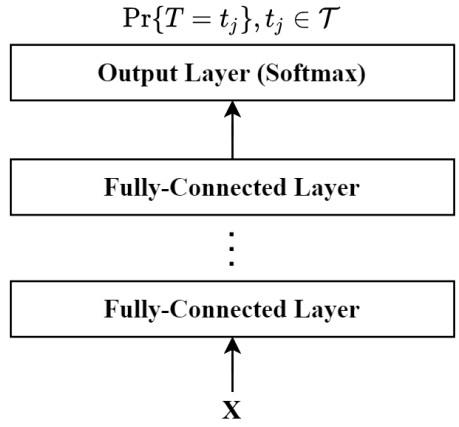

536

537                    Figure 1 The architecture of the DNN model

538

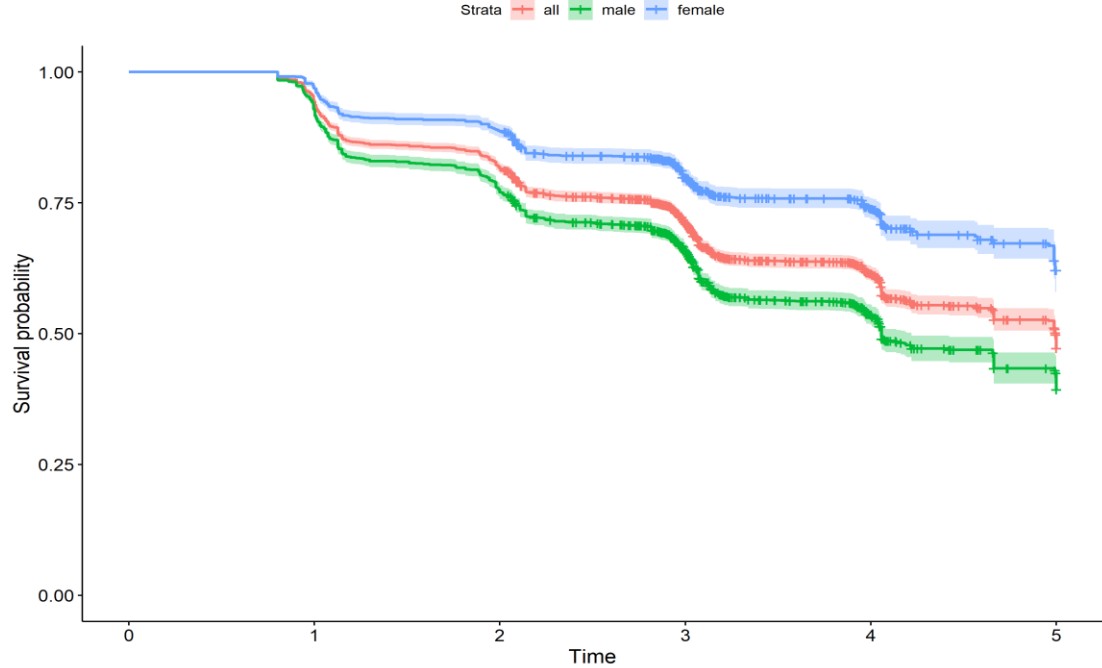

539

540        Figure 2 Kaplan-Meier survival estimates comparing male with female participants

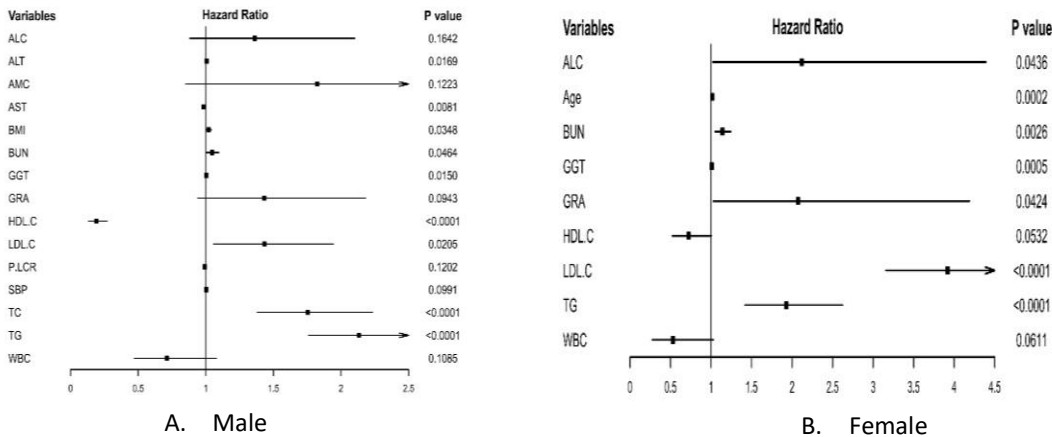

A.    Male

B.    Female

541

542        Figure 3 Cox Proportional Hazard Model for Predicting Dyslipidemia in Male (A) and Female (B)

543

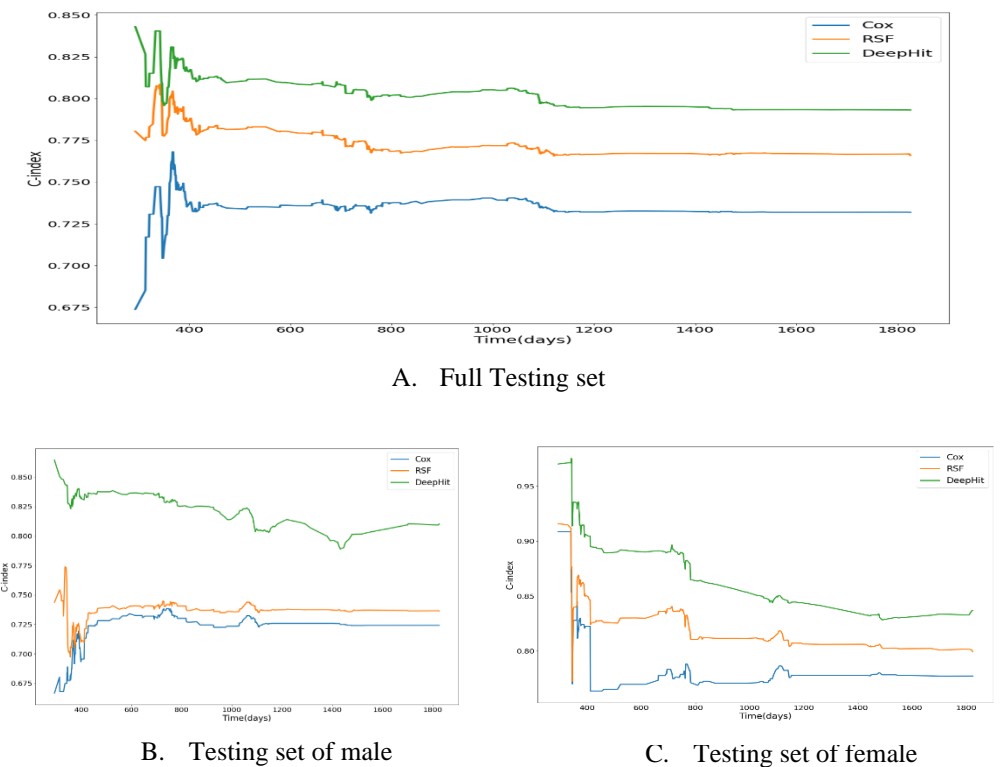

A.    Full Testing set

B.    Testing set of male

C.    Testing set of female

544

545    Figure 4 Comparison of $C^{td}$-index performance in Cox, RSF and DNN Models based on full

546    testing set (A); by the male (B) and female (C)