# OpenReview forum: "$$Predict the Risk of Dyslipidemia via Deep Neural Networks for Survival Data$$"
_KDD.org/2024/Workshop/AIDSH — KDD-AIDSH 2024 Poster_

### Official Review · Reviewer_akey · 2024-06-08
**A novel and interesting research focuses on dyslipidemia risk prediction with DNN, but this manuscript should be improved.**

**Rating:** 6
**Confidence:** 4

**Review:**

Authors try to adopt DNN to predict the risk of dyslipidemia, and  it is a good research manuscript. Especially, it involves more than 6000 subjects for this study, and it will influence other studies in the future. There are suggestions given to authors.\
1.**Background**: Some numbers, such as 101/1000, should be written as a percentage." In this translation, the fraction 101/1000 is mentioned as an example of a number that should be expressed in percentage form, which would be 10.1%.\
2.The **abbreviation** of Cox Proportional Hazards is different in this version, including COX and CPH. The same word should be expressed the same way in the manuscript. At the same time, authors is expected to carefully review this paper to avoid typos.\
3.**Figures** and tables should be placed inside the text, the current version needs to be improved, and the layout of references also needs to be optimized. Besides, the image should  be with high-resolution, and the fontsize needs to be changed. The font of the current version of the image appears to be stretched, which might provide a bad experience to read this manuscript.\
4.The **details of DNN** is unclear in this manuscript, and authors should provide **a novel figure 1** including more information.I can't even clearly know how many layers exist in the neural network model, I do not know the number of neurons and the activation function of the hidden layer. I do not know whether the normalization and dropout are used in this model. \
5.The reason for choosing **softmax** in regression model should be given, as softmax mainly works for classification. Since the sum of the output vector is 1, does it meet the requirement of your task?\
6.**Table 1** shows the gender difference, but I think the key point is the dyslipidemia. Could you provide the statistical data to demonstrate the difference between the health and unhealth population?\
7.**Since Lee et al[19] proposes DeepHit, authors need to demonstrate the difference and improvement.** Is it  completely same?  It is noticed that the evaluation metric and research process is similar to the previous study, like loss function.

---

### Official Review · Reviewer_PQoo · 2024-06-12

**Rating:** 4
**Confidence:** 4

**Review:**

This paper conducts risk prediction of dyslipidemia via DNN and reveals the value of application in real-world clinical scenarios. The paper is overall fluent and easy to follow. However, there are still several weaknesses and problems to be solved, most of which are technical problems:
1. The DNN method is not well introduced. There is only a figure that shows the architecture, without reporting the hyperparameters such as the number of FC layers, the hidden dimension size, etc. Moreover, the optimizer of this algorithm is also important, I am curious whether you are using Adam or SGD, or anything else.
2. The loss function is unclear. There is no formal introduction about the optimization objective of the DNN network.
3. The data used is not well formulated. It is unclear that how the data is inputed to the DNN. Is it multivariate time-series? If so, how do you deal with the time steps? By concatenating every timestep and feature to a single vector? There is no explicit justification.
4. The method is relatively old and without novelty. It seems that the method directly inherits DeepHit without any modification. However, this is a relatively old method, and is not good at dealing with time-series data. Simply applying an existing and old algorithm may lack some novelty technically. Consider some more recent methods like SAFARI [1] or propose some technical innovations.

[1] Ma X, Wang Y, Chu X, et al. Patient health representation learning via correlational sparse prior of medical features[J]. IEEE Transactions on Knowledge and Data Engineering, 2022.

---

### Decision · Program_Chairs · 2024-06-28

Accept (Poster)